# Case Report: A Sudden Thyroid-Related Death of a 15-Year-Old Girl

**DOI:** 10.3390/diagnostics14090905

**Published:** 2024-04-26

**Authors:** Kálmán Rácz, Gábor Simon, Andrea Kurucz, Gergő Tamás Harsányi, Miklós Török, László Tamás Herczeg, Péter Attila Gergely

**Affiliations:** 1Clinical Center, Department of Forensic Medicine, University of Debrecen, 4032 Debrecen, Hungary; raczkalman@hotmail.com (K.R.); herczeg.l.t@gmail.com (L.T.H.); gergely.peter@med.unideb.hu (P.A.G.); 2Medical School, Department of Forensic Medicine, University of Pécs, 7624 Pécs, Hungary; 3Clinical Center, Department of Cardiology and Cardiac Surgery, University of Debrecen, 4032 Debrecen, Hungary; andreakurucz12@gmail.com; 4Pathology Department, Szabolcs-Szatmár-Bereg County Teaching Hospital, 4400 Nyíregyháza, Hungary; harsanyi.tamas.gergo@gmail.com; 5Clinical Center, Department of Pathology, Kenézy Gyula Campus, University of Debrecen, 4032 Debrecen, Hungary; torokmi77@gmail.com

**Keywords:** thyroid storm, thyroid gland histology, elevated thyroid hormone level, thyrotoxic crisis, sudden death

## Abstract

A 15-year-old young girl was found dead at home. There were no indications of any intervention or the application of force. On the previous day, she was admitted to hospital because of palpitations, fatigue, a headache, and a swollen neck. During a physical examination, a swollen thyroid gland and tachycardia were found. In the family history, her mother had thyroid disease. According to the laboratory values, she had elevated thyroid hormone levels. After administration of beta-blockers, the patient was discharged and died at home during the night. The parents denounced the hospital for medical malpractice; therefore, a Forensic Autopsy was performed. Based on the available clinical data, the autopsy, histological and toxicological results, the cause of death was stated as multiorgan failure due to disseminated intravascular coagulation (DIC) caused by the autoimmune Graves disease. The forensic assessment of the case does not reveal medical malpractice. Post-mortem diagnoses of thyroid disorders in cases of sudden death can be challenging. However, as the reported case illustrates, the diagnosis could be established after a detailed evaluation of antemortem clinical data, autopsy results, histology, and a toxicological examination.

## 1. Introduction

Sudden death due to thyroid disorders is a rare occurrence in forensic practice. For forensic pathologists, the main difficulty is noticing the indications of a potential sudden thyroid-related death and, even more, proving it, since the analysis of postmortem thyroid-stimulating hormone (TSH) levels is generally not part of everyday practice. 

Thyroid storm is an endocrine emergency—it has a high mortality rate (10–30%) [1]—as a result of an increased response to thyroid hormones, usually based on Graves disease. Other conditions, such as autoimmune thyroiditis and excessive extrathyroidal secretion of thyroid hormones, i.e., adenomas, can also lead to this pathologic process [2]. The exact incidence is hard to estimate as it is a rare condition with unspecified criteria and a lack of specific laboratory findings. However, studies are reporting the occurrence of thyroid storms between 1% and 10% among hospital admissions for thyrotoxicosis [1,3]. Patients usually feel intensified symptoms of hyperthyroidism, including tachycardia, fever, sweating, and palpitations with various signs of multiorgan failure (MOF) [4], e.g., with exercise intolerance and dyspnea as signs of heart failure [5], agitation and delirium as central nervous system (CNS) failure [6], nausea and vomiting as gastrointestinal tract failure, hepatomegaly, and liver dysfunction with progression to jaundice [7]. Unusual signs of thyrotoxic crisis can be acute abdomen, hypoglycemia, lactic acidosis, and disseminated intravascular coagulation with multiple thrombotic events [7,8]. 

Here, we report the case of the unexpected death of a 15-year-old girl suffering from thyrotoxicosis.

## 2. Case Report

A 15-year-old girl was admitted to the emergency department of a pediatric clinic on a Friday. She complained of weakness and fatigue, which started two months ago, and felt shortness of breath after 100 m running. She had experienced forehead pain, sometimes with nausea, three times a week for three months. She experienced amenorrhea in the last four months. She noticed neck swelling the previous month. She also complained of permanent joint pain; therefore, she occasionally took painkillers. The family history was positive for goiter (her mother). 

A physical examination revealed a swollen, stiff thyroid region. Her heart rate was between 103 and 130 bpm, and blood pressure was between 148/81 and 155/88 mmHg in the hospital. The electrocardiogram (ECG) showed normal sinus rhythm. Laboratory testing, chest X-ray, and neck ultrasound were also performed as part of the medical check-up. The ultrasound examination showed a diffusely swollen, hypoechoic thyroid gland with an 8 mm nodule in the right lobe.

Se TSH, fT3, and fT4 values were available by the end of the medical check-up, and they represented hyperthyroidism. Anti-TG and Anti-TPO values became available two days later, only after the death of the girl, showing autoimmune thyroiditis. Other laboratory parameters were within the normal range (Table 1).

Propranolol was prescribed (40 mg 2 × 1 daily) for her to control the tachycardia, and the patient was discharged at 15:32 with an endocrinological referral three days later. She was found dead by her parents the next morning. CPR was not attempted. The parents could not present any information about the condition of the victim after she went to sleep in the evening. Post-mortem signs were not recorded; thus, the exact time of death could not be established.

The parents denounced the doctor for suspicion of medical malpractice, stating that the prescribed propranolol was responsible for the death of the victim.

### 2.1. Autopsy Findings

A forensic autopsy was performed according to the Recommendation No.R (99)3 of the Council of Europe [9]. During the external examination, apart from the swollen thyroid region (goiter), no physical alterations or injuries were found on the body—not even the external signs of hyperthyroidism (such as proptosis, thyroid dermopathy, or acropachy) [2]. The victim was 167 cm in length and weighed 62.8 kg (BMI: 22.5). Cerebral hemispheres showed generalized edema without any macroscopically visible focal pathology. (Figure 1A). The thyroid gland was enlarged, with a size of 8 × 4 cm on both sides. A pedunculated tumor with 2 cm diameter was found at the lower pole of the right lobe. The cut surface of the thyroid gland was normal. (Figure 1B). The thymus was enlarged at 7 × 7 × 2 cm in size. Macroscopically, the thymus and the thyroid gland were attached to each other. The right ventricle of the heart was dilated, and petechial hemorrhages were found on the epicardium (Figure 1B). During the autopsy of the heart, the coronaries and the chambers of the heart were opened using transversal cuts [10]. The left ventricle wall thickness measured at mid-cavity level was 11 mm, with a septum thickness of 12 mm, and the right ventricle thickness was 2 mm at the same level. No pathological condition was found in the heart apart from the above-mentioned epicardial petechial bleedings, and the coronaries were free from atherosclerosis or thrombosis. The lungs were slightly overinflated, but no other macroscopically visible alteration was observed. The cut surfaces were dry, and no fluid could be pressed out. No emboli were found in the first and secondary branches of the pulmonary arteries. The spleen was macroscopically hypovolemic. A focal parenchymal hemorrhage was observed in the parenchyma (Figure 1C), and also mucosal bleeding was seen in the proximal part of the small intestine with focally blood-stained content (Figure 1D). The ovaries were normal-sized, but multiple cysts of 0.5 cm could be seen, filled with blackish–red content (Figure 1E). Examination of the superficial and deep veins of the lower extremities revealed no presence of thrombosis, similar to the examination of the venous system of the small pelvis. Figure 2 illustrates the neck-lung-heart complex (Figure 2).

The organ weights were as follows: brain 1227 g, heart 303 g, spleen 380 g, liver 1511 g, kidney 321 g, thyroid gland 32 g. Histological samples were collected from the brain, (prefrontal cortex, cerebellum), heart (left ventricle (anterior lateral and posterior wall) septum, right ventricle (anterior, lateral and posterior wall)), sinoatrial node and Koch’s triangle (AV node), lung, thyroid gland, thymus, spleen, small intestine, pancreas, ovarium, kidneys, and liver. Toxicological samples were collected from the femoral vein (whole blood) and bladder (urine). Performing post-mortem biochemistry was not available.

### 2.2. Histological Findings

All samples were fixed with 9% buffered formalin and stained with hematoxylin and eosin (HE).

The thyroid glands showed mild autolysis. The follicular epithel cells were swollen and colloid could not be seen in the follicules. In a circumscribed area, in a fibrotic capsule normo-, micro-, and macrofollicules could be seen. In some areas, there were variable patchy lymphoid infiltrates in the stroma, but these did not form lymphoid follicules (Figure 3).

The cortical and medullary compartments of the thymus were hyperplastic (Figure 4A). A venous thrombus surrounded by an initial hemorrhagic infarct was detected in the pancreas (Figure 4B). The alveolar structure of the lung was maintained, but an occluding thrombus could be seen in a branch of the pulmonary artery (Figure 4C). The spleen showed signs of congestion (Figure 4D). Subcapsular, multiple cystic follicles could be seen in the ovarial stroma. Diagnosis: polycystic ovary disease (Figure 4E). Severe congestion was observed in the mucosal vessels of the small intestine (Figure 4F). The histology of the heart and kidneys was unremarkable. Apart from the lung and pancreas, no thrombosis or emboli were found in the samples from other organs. 

### 2.3. Toxicological Results

A toxicological analysis was carried out using liquid chromatography (Agilent LC 1200 Series/DAD, Santa Clara, CA, USA) and a gas-chromatography system (Agilent 7890 A—5975C MSD, Santa Clara, CA, USA). During the toxicological analysis, naproxen was detected in the blood in 40.67 ug/mL concentration, but propranolol could not be detected (LOD: 10 ng/mL).

## 3. Discussion

Thyrotoxicosis can be caused by several pathologies, such as Graves disease, toxic multinodular adenoma (TNMA), toxic adenoma, TSH-producing pituitary adenomas, resistance to thyroid hormones, trophoblastic disease, thyroiditis, follicular thyroid cancer, or iatrogenic effect (levothyroxine overdose) [11,12]. A thyroid storm is an endocrine emergency requiring prompt recognition and treatment, with the clinical signs of fever, tachycardia, nausea, confusion, altered sensorium, and gastroenterological hyperactivity [Shahid]. However, because of the significant overlap between these symptoms and other acute clinical conditions, objective diagnostic systems, such as Burch and Wartofsky score (BWSs) [13] or Akamizu criteria (Ak) [6], are advisable to be used in setting the diagnosis of thyroid storm [10]. Burch and Wartofsky’s scoring system assesses body temperature, gastrointestinal–hepatic dysfunction, cardiovascular symptoms, central nervous system disturbance, and precipitant history. A score above 45 indicates a thyroid storm; storm is impending between 25 and 44 and unlikely to be below 25 [14].

The general medical check-up in the reported case—containing detailed medical history, physical examination, electrocardiogram (ECG), chest X-ray, neck ultrasound, and laboratory testing—revealed thyrotoxicosis and autoimmune thyroiditis. As the results of the medical check-up did not fulfill the criteria of thyroid storm (Burch and Wartofsky score: 20–40), emergency hospitalization was not justifiable. Propranolol was prescribed to the patient—as it is a widely used drug against hyperthyroidism-induced tachycardia [15]—and a thorough endocrinological examination was organized for the following Monday, three days later. However, the examination could not be carried out because the girl suddenly died at home on the night of the presentation. 

The diagnosis of thyrotoxicosis was based on the clinical data (clinical presentation, family history of thyroid problems, and blood test results), with autopsy and histology revealing the cause of it. The availability of clinical data was essential for detecting thyrotoxicosis, since its clinical signs could not be observed during autopsy, and other signs were also lacking. If no clinical data are available, then information on symptoms could be acquired from relatives. If no clinical data about thyroid function are available, then postmortem thyroid function testing can be advised; however, its interpretation is complicated by postmortem changes that lead to alterations of the level of hormones. The literature is contradictory on this matter, and even the postmortem vitreous levels of these hormones seem unreliable [16,17]. The lack of external signs suggests an acute onset of the disease (which corresponds well with the short medical history). Cerebral edema, pancreas and intestinal bleeding, pulmonary arterial thrombosis and consequential right heart ventricle dilation can be manifestations of disseminated intravascular coagulation (DIC) and MOF caused by thyrotoxicosis [18]. 

According to the scientific literature, the presence of splenomegaly and thymic hyperplasia are well-known alterations associated with Graves disease. However, their underlying mechanisms are not entirely elucidated [19,20]. 

Polycystic ovary syndrome (PCOS) is a fairly common disorder and can cause severe problems, including menstrual abnormalities, anovulation, and infertility [21]. It is thought to be evoked based on visceral obesity and insulin resistance [22]. However, recently, more and more studies have supported the theory that other mechanisms, such as insulin resistance, might be involved in PCOS development. Previous reports claimed that a significant portion of women with PCOS had quite normal insulin sensitivity, were lean and did not respond to metformin therapy [23,24]. Some studies have raised the idea that autoimmunity might play a role in the pathogenesis of PCOS [25]. Antibodies against smooth muscles and anti-nuclear antibodies have been discovered to react with ovarian tissue [26]. There are also several publications suggesting a relationship between PCOS and autoimmune thyroid diseases, i.e., Hashimoto thyroiditis or Graves’ disease [27]. Janssen et al. recently compared 175 PCOS patients and 168 healthy women, and they found a significantly higher level of anti-thyroid antibodies in the PCOS group than in the control group (27% versus 8.3%) [28]. Following this report, a rising number of studies proved that there might be a connection between PCOS and a higher prevalence of thyroid diseases [29,30,31,32]. 

The evaluation of the autopsy results of cases of sudden death can be challenging. When determining the cause of death, one must consider all possibilities [33]. Graves disease was proven based on previous clinical data and autopsy findings. The victim also showed cardiac symptoms prior to her death, so the possibility of sudden cardiac death has also been examined. The detailed autopsy showed no macro or microscopic sign of heart disease (also, heart size and myocardial wall thickness were in the normal range [34]). Graves disease, however, could explain the cardiac symptoms. The autopsy findings can be explained by Graves’s disease (as its complications) or could not play a causative role in the death of the victim (like PCOS). Toxicological analysis is also essential in all cases of sudden death, but it did not reveal the presence of any substance that could contribute to the death of the victim.

Based on the clinical data, macroscopic and microscopic results of the autopsy and the toxicological analysis, thyrotoxicosis based on Graves disease was determined as the natural cause of death. The exact mechanism of death, however, could not be established, as tachyarrhythmia or high-output cardiac failure could be considered (especially as the recommended propranolol was not used), but the presence of petechial bleedings, microthrombi of lungs and pancreas, and small intestine bleeding points toward DIC and multi-organ failure. The cause of death was stated as multiorgan failure due to DIC with pulmonary embolism caused by the autoimmune Graves disease. The toxicology report ruled out the possibility of alleged propranolol overdose and pointed out that despite the prescription there was non-compliance.

The hospital treatment of the victim was appropriate to her condition and no further urgent (acute) examinations and treatment had to be completed, and a medical check-up was planned for the next weekday (Monday). Therefore, no suspicion of criminal responsibility was found from the hospital staff. The police decided also not to pursue a legal case against the parents.

## 4. Discussion

Thyrotoxic storm should be considered in cases of sudden death, particularly of females. Post-mortem diagnosing of thyroid disorders in cases of sudden death can be challenging. Detailed evaluation of ante-mortem clinical data, autopsy results, histology, and a toxicological examination, however, can help to establish the diagnosis. In the absence of clinical information, the signs may be subtle (even more so in acute cases where there has not been development of weight loss and eye symptoms); performing post-mortem hormone tests can be advised in these cases if suspicion is raised. The case is also an excellent example of PCOS and autoimmune thyroid disease coincidence that supports the idea of autoimmunity in the pathogenesis of PCOS. 

## Figures and Tables

**Figure 1 diagnostics-14-00905-f001:**
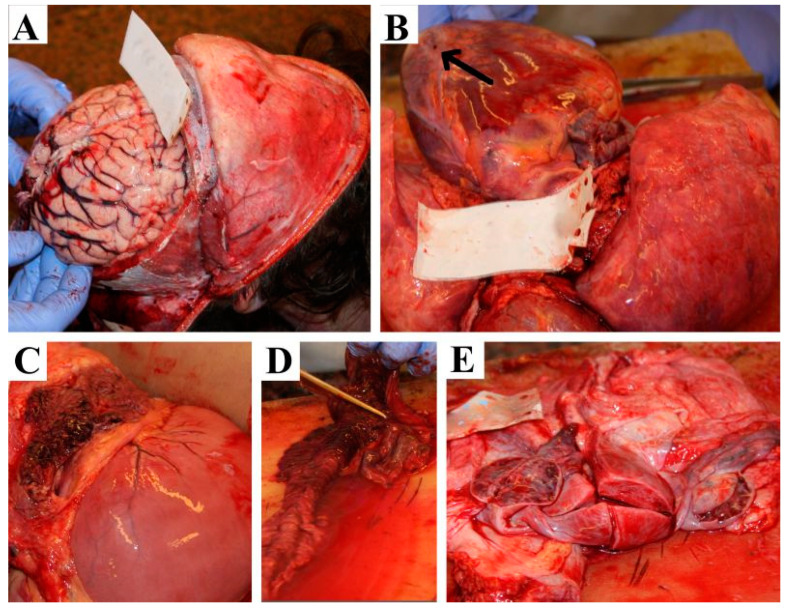
Autopsy findings. (**A**) Brain; (**B**) heart with petechial bleedings (arrow); (**C**) pancreatic haemorrhage spleen; (**D**) small intestine hemorrhage; (**E**) ovarian cysts.

**Figure 2 diagnostics-14-00905-f002:**
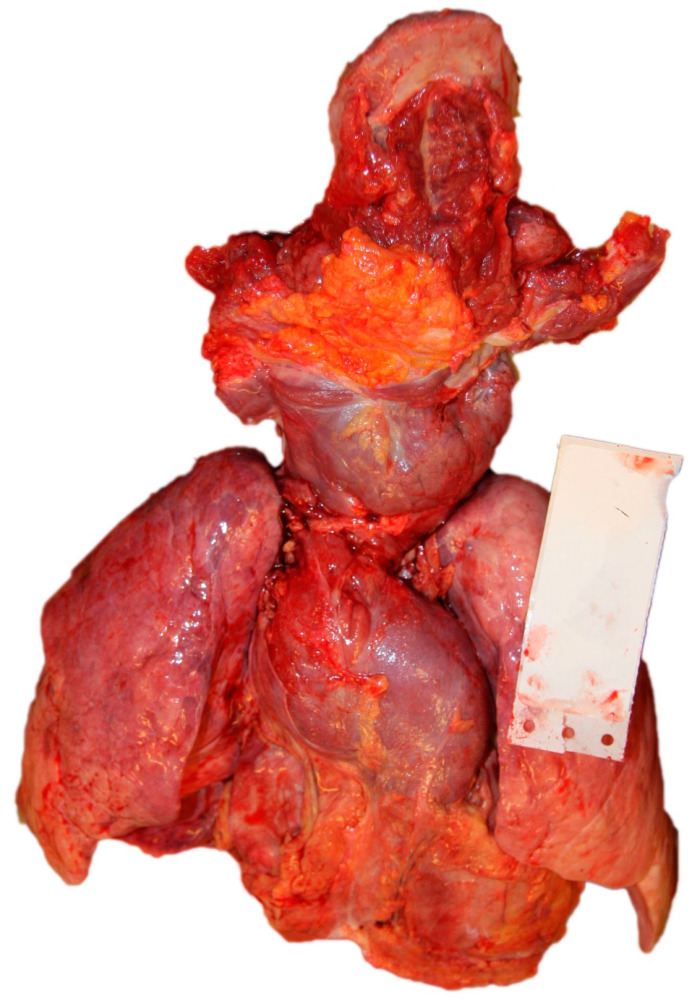
Autopsy findings (neck, lung, and heart complex).

**Figure 3 diagnostics-14-00905-f003:**
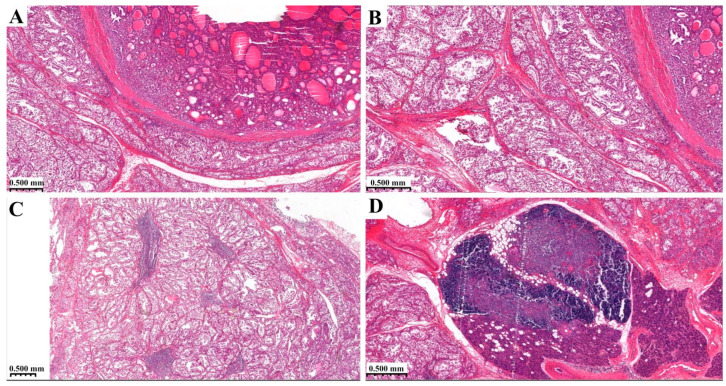
Histological findings of thyroid gland (HE staining). (**A**,**B**) normo-, micro- and macrofollicules in fibrotic capsule, (**C**) lymphoid infiltrates in the stroma, (**D**) attachment of thyroid gland and thymus.

**Figure 4 diagnostics-14-00905-f004:**
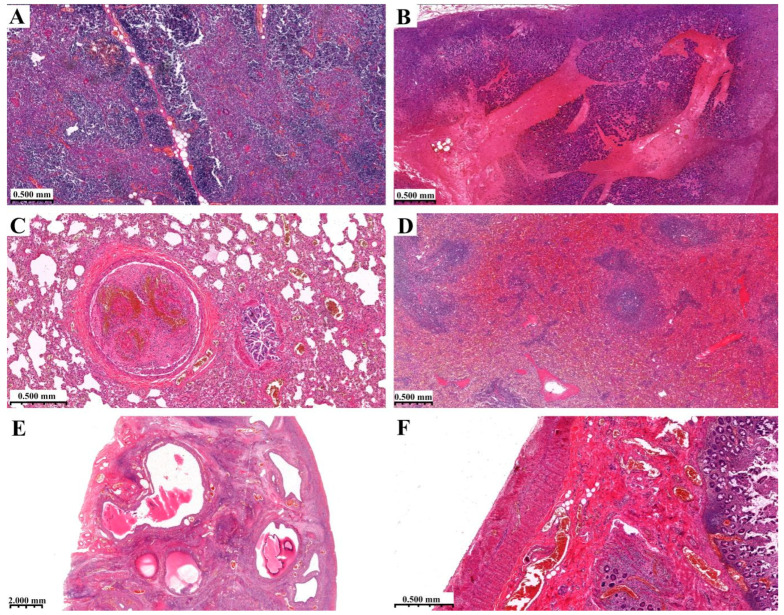
Histological findings (HE staining). (**A**) Thymus, (**B**) pancreas, (**C**) lung (thrombus), (**D**) spleen, (**E**) ovarium, (**F**) small intestine.

**Table 1 diagnostics-14-00905-t001:** Laboratory findings. CRP: C-reactive protein, T3: tri-iodine-thyronin, T4: tetra-iodine-thyronin, anti-TG: anti-thyroglobulin, anti-TPO: anti-thyroid peroxidase. ^1^ Anti-TG and anti-TPO results were available after the death of the girl.

Parameter	Value	Unit	Normal Range	
			Min	Max
sCRP	0.4	mg/L	0.0	3.0
sTSH supersensitiv	<0.008	µIU/mL	0.510	4.940
sFree T3	>30.80	pmol/L	3.50	6.50
sFree T4	88.82	pmol/L	11.50	22.70
Anti-TG ^1^	2479	IU/mL	0.00	<60.00
Anti-TPO ^1^	135.9	U/mL	0.00	<16.00

## Data Availability

All data are contained within the article.

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
