# Peer review of "Case Report: A Sudden Thyroid-Related Death of a 15-Year-Old Girl"

_diagnostics, 2024, doi:10.3390/diagnostics14090905_

Round 1
Reviewer 1 Report
Comments and Suggestions for Authors
This paper deals with a case of sudden death in a young girl, previously admitted in hospital for tachicardia and high blood pressure. She died during the night, with no anticipating symptoms. An autopsy report has been showed.
I have some comments. The article must be revised, author did not provide strong evidence to support their diagnosis.
First of all, the girl showed cardiac signs before death. So, a good forensic evaluation MUST departed from Basso et al (2017) guidelines:
The role of autopsy is «to establish or consider:
whether death is attributable to a cardiac disease or to other causes of SD;
the nature of the cardiac disease, and whether the mechanism was arrhythmic or mechanical;
whether the condition causing SD may be inherited, requiring screening and counseling of the next of kin
the possibility of toxic or illicit drug abuse, trauma, and other unnatural causes;
the role of third persons in the death.»
There are several errors in your procedure:
1. The lack of central blood sample. Why did you not sample central blood? It represent the only way to establish current use of a substance of abuse.
2. Heart sampling. You sampled «heart (left ventricle, right ventricle, sinoatrial node and Koch’s triangle (AV node)». The protocol of heart examination is completely wrong.
Please provide the description of Basso C, et al; Association for European Cardiovascular Pathology. Guidelines for autopsy investigation of sudden cardiac death: 2017 update from the Association for European Cardiovascular Pathology. Virchows Arch. 2017 Dec;471(6):691-705. doi: 10.1007/s00428-017-2221-0. Epub 2017 Sep 9. PMID: 28889247; PMCID: PMC5711979.
First of all, «make multiple transverse cuts at 3-mm intervals along the course of the main epicardial arteries»

Than, provide a complete transverse (short-axis) cut of the heart at the mid-ventricular level and then further parallel transverse slices of ventricles at 1 cm intervals towards the apex.

The standard histologic examination of the heart:

Dirty and unclear autopsy. The stain on the heart is a hypostasis stain, or a Tardieu stain, not a petechia.
The pancreas and small intestine are moderately infiltrated with blood, but not hemorrhagic. Kidney sample shows any embolus?
In my opinion, evidence of a blood mark on duodenum and pancreas is related to a pancreatitis complicated by a systemic thromboembolic mechanism.
The cause of death appears weak. I think, the lung embolus must be better considered. The deep venous circulation of the lower limbs needed to be evaluated. The lungs appear hyper-expanded. The appropriate study of the heart-lung block involves the opening of the pulmonary artery up to its secondary branches.
The seriation of causes of death is:
Multiorgan failure with respiratory failure due to DIC with pulmonary embolism, both causated by an autoimmune polyendocrinopathy.
Author Response
On behalf of all the authors, I would like to thank you for reviewing our
manuscript entitled “A sudden thyroid death of a 15-year-old-girl.” Thank you for your work, especially not only for highlighting the mistakes but also for suggesting the necessary corrections. We hope you will find our corrected manuscript worth publishing after this revision.
The changes were marked with red letters in the manuscript file.
This paper deals with a case of sudden death in a young girl, previously admitted in hospital for tachicardia and high blood pressure. She died during the night, with no anticipating symptoms. An autopsy report has been showed. I have some comments. The article must be revised, author did not provide strong evidence to support their diagnosis
Thank you for your thorough reading and review of our manuscript. We made the corrections you suggested. Below, you can find detailed answers to your comments.
First of all, the girl showed cardiac signs before death. So, a good forensic evaluation MUST departed from Basso et al (2017) guidelines:
The role of autopsy is «to establish or consider:
whether death is attributable to a cardiac disease or to other causes of SD;
the nature of the cardiac disease, and whether the mechanism was arrhythmic or mechanical;
whether the condition causing SD may be inherited, requiring screening and counseling of the next of kin
the possibility of toxic or illicit drug abuse, trauma, and other unnatural causes;
the role of third persons in the death.»
1.
Thank you for this important observation. We agree that Forensic Evaluation of sudden death should follow the recommended guidelines, and all possibilities must be assessed. We inserted new references for this reason and discussed it in a new paragraph in the discussion section (we think that further detail is not necessary in a case report – as the reference would detail it for those who want to be engrossed in this topic).
The lack of central blood sample. Why did you not sample central blood? It represent the only way to establish current use of a substance of abuse.
2.
As a general rule of Forensic Pathology, the blood sample for toxicological analysis should be collected from a peripheral vein, not from the heart. This is the rule set by the current (2023) protocol of the author’s country, and also by the Recommendation No.R (99)3 of the Council of Europe (Title: „On the harmonization of medico-legal autopsy rules”): „in all autopsies, the basic sampling scheme includes specimen from the main organs from histology and peripherial blood sampling „
The scientific literature also suggests the same, below you can find some examples:
- Øiestad ÅML, Karinen R, Rogde S, et al. Comparative Study of Postmortem Concentrations of Antidepressants in Several Different Matrices. J Anal Toxicol. 2018;42(7):446-458. doi:10.1093/jat/bky030: „Peripheral blood (PB) is considered to be the golden standard for measuring postmortem drug concentrations.”
- Hilberg T, Rogde S, Mørland J. Postmortem drug redistribution--human cases related to results in experimental animals. J Forensic Sci. 1999;44(1):3-9: „Femoral blood is widely accepted as the most reliable postmortem specimen for drug analysis in forensic toxicology.”
In conclusion, the peripheral vein is the recommended source of blood samples in the case of Forensic Autopsies, as it is the most reliable source of blood for toxicological analysis of substance use.
Heart sampling. You sampled «heart (left ventricle, right ventricle, sinoatrial node and Koch’s triangle (AV node)». The protocol of heart examination is completely wrong.
Please provide the description of Basso C, et al; Association for European Cardiovascular Pathology. Guidelines for autopsy investigation of sudden cardiac death: 2017 update from the Association for European Cardiovascular Pathology. Virchows Arch. 2017 Dec;471(6):691-705. doi: 10.1007/s00428-017-2221-0. Epub 2017 Sep 9. PMID: 28889247; PMCID: PMC5711979.
3.
Your suggestion of the autopsy technique is absolutely valid.
However, we employed an autopsy techinique fulflling the main recommendations of that protocoll: the heart was open with transverse cuts starting from the apex and going upward thus creating three transverse slice; coronaries were opened by transverse cuts, and multiple histological samples were taken from both ventricules (from all sides of the heart).
We did not detail the above-described procedure in our manuscript because of the unremarkable findings, and we thought that describing the autopsy technique is not information that should be described in a case report (we kept it in our mind to keep the manuscript relatively short).
But considering your suggestion, we agree with you that clarifying the autopsy technique should help understanding, and helps substantiate our conclusions. So, upon your suggestion, we added information about how the autopsy of the heart was made and also added a reference to the protocol.
Dirty and unclear autopsy. The stain on the heart is a hypostasis stain, or a Tardieu stain, not a petechia.
4.
The questions emerging about autopsy technique were answered above (and corrected in the manuscript). The staining on the heart is maybe not clear in the photograph (as it is a really small finding – we regret that we did not have a better picture of it)), but it was a petechia without a doubt.
The pancreas and small intestine are moderately infiltrated with blood, but not hemorrhagic.
5.
There is no gross picture of the pancreas, but the hemorrhage was just focal and not diffuse. Besides the moderate autolysis, the histology of the pancreas showed arteries and venous thrombi with initial hemorrhagic infarct due to DIC (this information we missed out from the original manuscript). The zonal distribution of hemorrhage around the thrombi makes the diagnosis of pancreatitis doubtful.
In the duodenum, not only were the vessels filled with blood, but the content was also bloodstained, which makes clear that it is not a passive congestion but rather focal bleeding (we included this information in our manuscript).
Kidney sample shows any embolus?
6.
No macroscopic or microscopic emboli were found in the kidneys. We included information about the negative kidney histology in the revised version.
In my opinion, evidence of a blood mark on duodenum and pancreas is related to a pancreatitis complicated by a systemic thromboembolic mechanism.
7.
Your opinion is absolutely valid based on the information in the original version of the manuscript because we unintentionally left out information; thus, the findings became misleading. Based on your observations, we noticed our mistake and corrected it in the revised version.
The cause of death appears weak. I think, the lung embolus must be better considered. The deep venous circulation of the lower limbs needed to be evaluated.
8.
The superficial and deep venous circulation of lower limbs and the venous plexus contained no thrombosis. Thank you for noticing that this information was missing from the manuscript. We have included it in the revised version.
The lungs appear hyper-expanded. The appropriate study of the heart-lung block involves the opening of the pulmonary artery up to its secondary branches.
9.
Yes, you observed well; the lungs were slightly hyper-expanded (overinflated). Based on your comment, we noticed that we left out the macroscopic description of the lungs, so we corrected this mistake. The lung arteries were opened to the secondary branches, but no macroscopically visible thrombosis was found (we included this information in the revised manuscript).
The seriation of causes of death is:
Multiorgan failure with respiratory failure due to DIC with pulmonary embolism, both causated by an autoimmune polyendocrinopathy.
10.
Your opinion regarding the seriation of causes of death is quite similar to ours, but – as we noticed upon your observation – it was a little confusing in our original manuscript. We corrected it.
Reviewer 2 Report
Comments and Suggestions for Authors
I have reviewed the manuscript "Case report: A sudden thyroid death of a 15-year-old girl", which concerns an interesting clinical case. The manuscript is well-written ad easy to follow. Please see my concerns below:
- my major concern is that reports on a case in which there seems to be judicial involvement because the parents of the patient denounced the hospital for medical malpractice. This is an issue of major warning to further proceed with publication of this case. I think the authors should provide the journal with appropriate information on this regard, and then probably evalute the need of providing further information on this matter within the manuscript publication.
- further information and clear statements on whether the authors consider that a different clinical conduct should have been undertaken before discharge on the presentation day is missing (e.g. the endocronilogical examination that was organized for monday could have been organized earlier, and whether it would have led to different outcomes, etc...).
- it was not immediately evident to me that this case report in focused on forensic interests. I suggest to make this more evident in the title and abstract.
- I would also suggest the authors to check on agreement with author guidelines because the manuscript seems somewhat too large for a case report to me.
Comments on the Quality of English LanguageI detected some typos.
Author Response
I have reviewed the manuscript "Case report: A sudden thyroid death of a 15-year-old girl", which concerns an interesting clinical case. The manuscript is well-written ad easy to follow. Please see my concerns below:
Dear Reviewer 2
Thank you for your support of our manuscript. Your concerns could be valid in general; you can find the answers below.
my major concern is that reports on a case in which there seems to be judicial involvement because the parents of the patient denounced the hospital for medical malpractice. This is an issue of major warning to further proceed with publication of this case. I think the authors should provide the journal with appropriate information on this regard, and then probably evalute the need of providing further information on this matter within the manuscript publication.
The police decided not to pursue a legal challenge against the parents. However, we have waited with the publication of this case until possible criminal responsibility is forfeited. We included this information to the informed consent statement.
- further information and clear statements on whether the authors consider that a different clinical conduct should have been undertaken before discharge on the presentation day is missing (e.g. the endocronilogical examination that was organized for monday could have been organized earlier, and whether it would have led to different outcomes, etc...).
Thank you for this remark; we included this information in the last paragraph of the discussion section.
It was not immediately evident to me that this case report in focused on forensic interests. I suggest to make this more evident in the title and abstract.
The abstract was expanded to make the forensic interests more evident.
I would also suggest the authors to check on agreement with author guidelines because the manuscript seems somewhat too large for a case report to me.
The journal „Diagnostics” is an online-only journal; it has no limitation to the length of a case report.
Round 2
Reviewer 1 Report
Comments and Suggestions for Authors
A good response to my revision is provided. Congrats
Author Response
Dear Reviewer 1!
Thank you for your support of our manuscript, your previous remark helped greatly to elevate the standard of our manuscript.
Thank you:
Authors
Reviewer 2 Report
Comments and Suggestions for Authors
In relation to my concern on ethics and legal liability, the authors added "The case report was published after all possible criminal responsibility forfeited". I wonder how could this be forfeited. Legal cases can change 25 years later... It seems an overstatement to me.
In relation to my concern regarding large of the manuscript for this type of paper the authors replied "The journal „Diagnostics” is an online-only journal; it has no limitation to the length of a case report". This does not seem to be in line with the Article Types section of MDPI journals, wherein a 2500 word limit is established.
https://www.mdpi.com/about/article_types
Comments on the Quality of English LanguageNone.
Author Response
Dear Reviewer!
Thank you for supporting our manuscript. Below, you can find the answers for your latest comments.
1.
The legal rules could be very different in different countries, so we absolutely understand the reason behind your remark regarding the forfeiture time.
In the author’s country, the forfeiture time is the same as the maximum possible imprisonment length to the given crime, but not less than five years.
See the Criminal Code of Hungary (https://thb.kormany.hu/download/a/46/11000/Btk_EN.pdf)
Section 26:
„(1) unless otherwise provided for by the Act on the Exclusion of Statutes of Limitation for Certain Crimes, prosecution is barred upon the lapse of time equal to the maximum penalty prescribed, or after not less than five years.”
In Hungarian law, medical malpractice in criminal law is considered as „Professional Misconduct” which is punisable with a maximum of five years imprisonment, if the patient dies:
Section 165:
„(1) Any person who engages in misconduct in the course of engaging in his profession, thus causing imminent danger to the life, bodily integrity or health of another person or persons by his failure to act with reasonable care, or causes bodily harm, is guilty of a misdemeanor punishable by imprisonment not exceeding one year.
(2) The penalty shall be:
- b) imprisonment between one to five years if the crime results in death;”
In the given case, the parents who did not gave the prescribed medicine to their child could be prosecuted only for involuntary/negligent homicide, which is also punishable with a maximum of 5 years imprisonment:
Sections 160:
„(4) Any person who commits negligent homicide is guilty of a misdemeanor punishable by imprisonment between one to five years.”
In case of civil legal issues, there is also a forfeiture time of 5 years (so the plaintiff has five years to file a lawsuit).
So, the forfeiture time isn the given case is 5 years.
2.
The MDPI editorial office informed us that there is no mandatory requirement (word count limit) in word count in the journal.
Regards:
Authors